# SPD: Sync-Point Drop for efficient tensor parallelism of Large Language Models

## Abstract

With the rapid expansion in the scale of large language models (LLMs), enabling efficient distributed inference across multiple computing units has become increasingly critical. However, communication overheads from frequent synchronization during distributed inference pose a significant challenge to achieve scalability and low latency. Therefore, we introduce a novel optimization technique, Sync-Point Drop (SPD) to reduce communication overheads in tensor parallelism by dropping synchronization on attention outputs. In detail, we first propose a block design that allows execution to proceed without communication through SPD. Second, we identify regions of communication redundancy, where dropping synchronization results in no loss of model performance. In addition, to extend SPD across all compute blocks, we employ a low-cost distillation, specifically targeting blocks giving quality degradation, to maximize accuracy recovery. For extreme blocks where performance degradation is severe, we introduce a new head grouping enhancements to amplify the distillation's recovery effect. The proposed methods effectively alleviate communication bottlenecks while minimizing accuracy degradation during LLM inference, offering a scalable solution for distributed environments.

## 1 Introduction

Large Language Models (LLMs) (Gunter et al., 2024; Brown et al., 2020; Bubeck et al., 2023; Touvron et al., 2023a;b; Zhang et al., 2022; Penedo et al., 2023; Jiang et al., 2023) have revolutionized the field of natural language processing (NLP), driving significant advancements in a wide range of applications, from machine translation and sentiment analysis to question answering and content generation. Their ability to understand and generate human-like text has opened new possibilities for both research and practical use. However, as these models grow in size and complexity, optimizing their performance becomes a crucial challenge, particularly in terms of latency.

One of the popular ways to offer low latency is to run LLM inference in distributed computing environments, notably using Tensor Parallelism (TP) (Shoeybi et al., 2019). TP enables distributed computations by sharding tensor operations into separated tracks or *blocks* that are then processed on parallel devices simultaneously. As a direct benefit, TP allows us to leverage device's quantity against on-board memory per-device, which is a good way to improve hardware utilization and accelerate inference for large scale models (Further discussion can be found in Section 3).

However, to maintain mathematical parity as on single-device inference, TP requires collective communication, or sync-points, throughout the model — these are communication barriers across all parallel devices to synchronize hidden representation tensors (the process is illustrated in Figure 1a). Because of its communicative nature, the overhead of sync-point is subjected to hardware systems, i.e. interconnect between devices, network connections between nodes, which can become a bottleneck on execution. Also, sync-points may be critical failure points in any distributed systems. As LLMs grow in size, one needs to use more compute devices, which necessitates more sync-points and further worsens system stability and inference latency. Therefore, optimizing sync-point would greatly improve the overall inference latency and system utilization.

Therefore, in this work, we propose **Sync-Point Drop** (SPD) a simple yet novel optimization technique with broad applications. Unlike the existing works which tried to improve the communication process itself (NVIDIA, 2019b; Jeaugey, 2019; Cheng et al., 2023) on system-level, SPD directly

removes sync-point on self attention output (as in Figure 1b) within the target budget. To enable SPD directly on decoder block, we first present a block design for SPD which minimizes the negative effect from lack of communication (see Figure 2). Second, we differently apply SPD to each blocks based on communication sensitivity, which is defined as the relative impact on downstream performance when all communications are dropped up to that point. From our observations we identified three regions of communication sensitivity: in-sensitive, sensitive, and extremely sensitive (see Figure 3). We show that in-sensitive blocks resulted in virtually **no degradation of performance in zero-shot** when SPD is applied for many popular open-source models at various sizes. For sensitive blocks, we propose block-to-block distillation which tunes the decoder block aware of SPD while applying significantly low cost weight update. For extremely sensitive blocks, which show large quality degradation even with block-to-block distillation, we introduce a novel SPD aware initialization for block-to-block distillation. Our experimental results show effective possibility of latency improvement with minimizing the accuracy degradation throughout diverse sizes of models. In summary, our contributions are:

- We propose novel block designs for sync-point drop that minimize information loss from lack of communication.

- We identify the sensitivity of each block within the model and classify them into three distinct categories, allowing for the application of tailored optimization strategies to each group based on their performance characteristics.

- Optimization strategies applied to blocks based on sensitivity (zero-shot dropping with no performance degradation, block-to-block distillation and self attention head-grouping enhancement leading to better accuracy/latency trade-off) enable a scalable solution for distributed environments.

- Empirical results on various datasets show sync-point drop can offer improved latency with minimum quality loss for all budgets in distributed environments.

## 2 RELATED WORKS

The inefficiency of large language models, which emerged with significant impact, has led to the development of numerous optimization techniques. Diverse techniques, i.e. quantization, pruning, emerged on model-level optimization with enormous redundancy of large language model. Quantization (Frantar et al., 2023; Xiao et al., 2023; Lin et al., 2024; Shao et al., 2024; Chee et al., 2023; Ashkboos et al., 2024) reduces the precision of model parameters, allowing for faster computations with minimal impact on performance. Pruning (Frantar & Alistarh, 2023; Sun et al., 2024; Liu et al., 2023; Xia et al., 2024) eliminates less critical parameters or neurons from the model, thereby reducing its size and computational complexity. Furthermore, in the aggressive scale of pruning, block skipping (Xia et al., 2024; Song et al., 2024), which involves bypassing certain blocks during inference based on block characteristics, further enhances efficiency by decreasing the number of operations aggressively required for prediction. This model-level optimizations are mostly focusing on compressive and computational effect that makes them more suitable for real-time and resource-constrained environments without sacrificing accuracy.

Also, due to the large amount of computation and memory occupation of the large language model, system-level optimizations (Shoeybi et al., 2019; Huang et al., 2019; Zhao et al., 2023; Aminabadi et al., 2022; Kwon et al., 2023) are explored for deployment. Different with model-level optimization, system-level optimization does not change any numerical values of a model. One of the distributed deployment technique, tensor parallelism (Shoeybi et al., 2019), enables fast serving of a model by parallel execution of a block into multiple devices. However, this technique requires large communication overhead between devices to keep numeric precision of execution flow. Considering the communication bottleneck of tensor parallelism, existing work also focus on improving the communication operation itself systematically, including ring-topology all-reduce (NVIDIA, 2019b) and tree-topology all-reduce (Jeaugey, 2019). Specifically for large models, ATP (Cheng et al., 2023) improves training efficiency by dynamically choosing the best parallel strategy.

In this paper, we achieve optimization benefits from the system perspective by leveraging model-level optimization (enabling SPD in the system with minimizing accuracy degradation in the model).

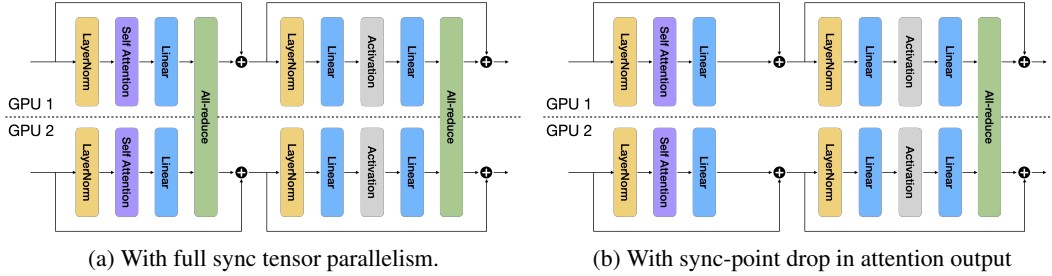

(a) With full sync tensor parallelism.    (b) With sync-point drop in attention output

Figure 1: Tensor parallelism applied on transformer decoder block (in 2-GPUs inference case).

# 3    PRELIMINARY: TENSOR PARALLELISM IN LARGE LANGUAGE MODELS

Tensor parallelism (Shoeybi et al., 2019) is a systematic computing technique on distributed environment used to accelerate large-scale language models. This is realized by partitioning individual weight tensors of a model across multiple devices. Instead of replicating entire model across GPUs (as in data parallelism), tensor parallelism (TP) divides each block's computation across multiple devices (as shown in Figure 1a), enabling the model to handle larger tensors that would otherwise exceed the memory capacity of a single GPU. This approach significantly improves the scalability and efficiency in both training and inference, particularly in large language models. However, realization of effective TP requires collective communication ('all-reduce' in Figure 1) between devices to synchronize and exchange partial computations, typically achieved via specialized interconnects such as NVLink and NVSwitch (NVIDIA, 2019a). To reduce communication overhead, careful coordination between computation and communication is crucial. In this paper, we introduce a novel method Sync-Point Drop (SPD) to eliminate the communication within each decoder block, aiming to alleviate communication bottlenecks during distributed inference.

Table 1 shows the latency of a decoder block inference with TP and SPD. To make the lowest bandwidth requirement between devices as possible, we measured latency with a block of LLaMA2-7B having an embedding dimension of 4096 and using an input with a batch size of one and a sequence length of one. In the single node case operating with 8-GPUs, SPD gives 7% latency gain (from 8.01ms to 7.45ms) compared to TP case. This becomes even larger in cross nodes environment. In double node case with 8-GPUs in each node, SPD gives 24% latency gain (from 28.78ms to 21.97ms) compared to TP case. Considering we utilize

Table 1: Latency of a decoder block of LLaMA2-7B model inference with tensor parallelism (TP) and sync-point drop (SPD) in diverse multi-gpu environments. (Latency metrics are measured in NVIDIA A100-80G gpus equipped with NVLink each other).

| Environment (#-node / #-gpus per node) | Per-block latency |
|---|---|
| 1-node / 8-gpus TP | 8.01 ms |
| 1-node / 8-gpus SPD | 7.45 ms |
| 2-node / 8-gpus TP | 28.78 ms |
| 2-node / 8-gpus SPD | 21.97 ms |

smallest specification to use lower amount of communication, the latency gap will be much bigger when larger batch size and sequence length are used with much larger models having a enormous embedding dimensions. This highlights the importance of SPD for efficient distributed inference.

# 4    SYNC-POINT DROP FOR EFFICIENT TENSOR PARALLELISM

Sync-point drop (SPD) simply removes the 'all-reduce' communication after self attention output as shown in Figure 1b. While the lack of communication harms numerical parity across all parallel devices, if the application is handled properly, it can be well applied with less quality degradation of the model. In this section, we propose several methods to enable keeping high quality model with lowering the communication overhead by SPD. First, in order to apply SPD, we introduce a novel block structure design that serves as the foundation block for the non-communicating structure with minimal quality degradation. Second, we propose proper strategy of applying SPD in a block-wise manner which can achieve benefits of low latency with minimizing the accuracy degradation.

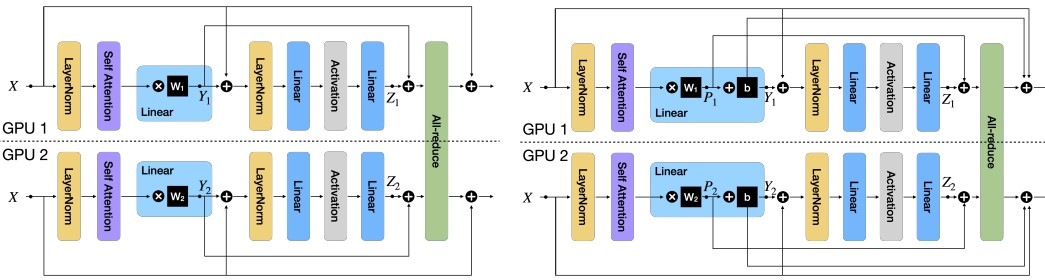

(a) Block design without bias in linear layer.      (b) Block design with bias in linear layer.

Figure 2: Decoder block structure with sync-point drop (in 2-GPUs inference case). '$W_i$' and '$b$' represent weight and bias of linear layer on each device ($i$). '$X$', '$Y_i$', '$Z_i$' and '$P_i$' denotes a hidden representation of each device ($i$) on '·' in the figure.

### 4.1 BLOCK DESIGN

If the synchronization of the self attention output is disrupted, the attention output will be divided into multiple outputs corresponding to the number of devices. This leads to design issues in two aspects of the transformer decoder block: the MLP input and the MLP output. Furthermore, these design issues vary depending on whether the output projection layer of self attention includes a bias term or not. In this section, we present a block design for SPD that minimizes information losses from lack of communication which lead to the accuracy degradation.

#### 4.1.1 DECODER BLOCK WITHOUT BIAS IN LINEAR LAYER

Figure 2a shows the SPD block design used without bias in linear layer. The most essential objective of block design is constructing the combination of connections which gives least numerical difference between TP and SPD.

**MLP input** The sync-point enables each parallel devices to capture the attention output from all the other devices. However, when the outputs from other devices are unavailable by elimination of sync-point, the only information the device ($i$) can utilize is its own attention output ($Y_i$). Therefore, to minimize the numerical difference compared to the case of all information available, residual connection ($X$) and attention output of own device ($Y_i$) are added and fed into MLP input ($X + Y_i$).

**MLP output** When the sync-point exists after attention output, the MLP input is utilized as residual connection added to MLP output. However, dropping the sync-point yields incomplete MLP input ($X + Y_i$) with lack of attention output from other devices. The desired block output is combination of block input ($X$), attention output from all devices ($\sum_i Y_i$) and MLP output from all devices ($\sum_i Z_i$). Therefore, we disassemble the original residual connection to block input residual ($X$) and attention output residual from a device ($Y_i$). Then, $Y_i$ forms a new type of residual connection which is added before sync operation. $X$ is added on the same point as original connection, after the sync operation which finally leads to complete form of output ($X + \sum_i Y_i + \sum_i Z_i$).

#### 4.1.2 DECODER BLOCK WITH BIAS IN LINEAR LAYER

In TP, each of the linear layers in self attention part of a block is parallelized in a different manner. The linear layers before self attention operation (query, key and value projection) are divided in a column-wise manner which enables the bias divided along same dimension. However, the linear layer after self attention operation (output projection) is parallelized in an orthogonal way, row-wise manner. The bias, a vector along the column dimension, therefore, can not be divided in the direction of the row. This requires new mechanism of the bias application on MLP input and output as shown in Figure 2b.

**MLP input** The difference from Section 4.1.1 is that indecomposable bias term ($b$) is included after weight multiplication. Following the most essential objective, least error in the MLP input compared to the result after communication, we use the partial weight multiplication result with the addition of bias ($Y_i = P_i + b$) and input residual connection ($X$) as MLP input ($X + P_i + b$).



Figure 3: Sync sensitivity identification process of a block ('TP' is for tensor parallelism and 'SPD' is for sync-point drop).

**Algorithm 1** Sync-point drop based on sensitivity

---

1: SPD: SYNC-POINT DROP
2: B2B: BLOCK-TO-BLOCK DISTILLATION
3: HG: ATTENTION HEAD GROUPING ENHANCEMENT
4: $Block \leftarrow$ *list of all decoder blocks in model*
5: $S \leftarrow$ *list of sensitivity measurement*
6: $B \leftarrow$ *block index list in ascending order of $S$*
7: $N_{spd} \leftarrow$ *Target budget: the number of blocks to SPD*
8: $\tau_1, \tau_2 \leftarrow$ *sensitivity thresholds*
9: **for** $i = 0$ **to** $N_{spd} - 1$ **do**
10:     **if** $S[B[i]] \leq \tau_1$ **then**   ▷ Section 4.2.2: in-sensitive
11:         $Block[B[i]] \leftarrow \text{SPD}(Block[B[i]])$
12:     **else if** $S[B[i]] \leq \tau_2$ **then**  ▷ Section 4.2.3: sensitive
13:         $Block[B[i]] \leftarrow \text{SPD}(\text{B2B}(Block[B[i]]))$
14:     **else**         ▷ Section 4.2.4: extremely sensitive
15:         $Block[B[i]] \leftarrow \text{SPD}(\text{B2B}(\text{HG}(Block[B[i]])))$
16:     **end if**
17: **end for**

---

**MLP output** Following Section 4.1.1, the original residual connection is disassembled to block input residual ($X$) and attention output residual ($Y_i$) from a device. Due to existence of bias, we further disassemble $Y_i$ to the result of the partial weight multiplication ($P_i$) and the bias ($b$). To make the bias not affected by communication, we place the bias residual add after the sync operation while adding the partial weight multiplication result before the sync operation. Finally, in a device, this makes the bias residual be added once on MLP output while the parallelized weight multiplication results form complete state through collective communication ($X + \sum_i P_i + b + \sum_i Z_i$).

## 4.2 SYNC-POINT DROP BASED ON BLOCK-WISE SENSITIVITY

While the lack of communication harms numerical parity across all parallel devices, we found that some blocks aren't affected a lot from SPD based on the designed block structure in Section 4.1. In fact, there is varying sensitivity to SPD depending on which block it is. In our proposed approach, we categorize transformer blocks based on their sensitivity: in-sensitive blocks, sensitive blocks and extremely sensitive blocks. This classification enables us to tailor our strategies on each block effectively, optimizing the inference processes block-by-block with less accuracy degradation.

### 4.2.1 BLOCK-WISE SYNC SENSITIVITY IDENTIFICATION

Figure 3 shows the overall sync sensitivity identification algorithm flow of a model in the parallel system setting. To identify the sensitivity of a block to SPD, we utilize perplexity metric by measuring relative impact of a block to performance (the difference between TP block and SPD block in Figure 3) as sensitivity measurement. For example, when we measure the sensitivity of $i$-th block to SPD, we apply SPD to all blocks starting from the $\{i + 1\}$-th block to the final block and measure the perplexity, while leaving the $i$-th block unchanged. Then we measure the perplexity by additionally modifying the system setting of $i$-th block to SPD. The difference in perplexity before and after applying SPD to $i$-th block is used as a measure of sensitivity. Here, we use calibration data obtained by sampling a small portion from the large training dataset. By progressive replacement of TP block to SPD block and measurement of quality degradation as relative perplexity difference, we can compare the sensitivity between blocks in entire model and classify the blocks to three sensitivity categories (in-sensitive blocks, sensitive blocks and extremely sensitive blocks).

Algorithm 1 shows the overall process of properly applying SPD in a block-wise manner with sync sensitivity. Based on the measured sensitivity value of blocks ($S$), we rank the blocks in an ascending order ($B$). Following this ranking of the sensitivity of each blocks, we apply SPD sequentially within the target number of blocks to optimize ($N_{spd}$). In the sequence, the processing of a block is classified based on a predefined threshold criterion ($\tau_1$ and $\tau_2$). This makes the blocks classified

into three sensitivity categories. Consequently, this classification allows us to implement individual strategies aimed at minimizing quality degradation according to the identified groups. In the following sections, we introduce the individual strategies, applied based on classification result.

### 4.2.2 IN-SENSITIVE BLOCKS: ZERO-SHOT DROPPING

In-sensitive blocks show minimal accuracy degradation with SPD. Therefore, within the targeted budget of communication optimization, we drop the sync-point of these blocks, prioritized over other types of blocks, in a zero-shot manner. Note that zero-shot dropping can give significant amount of benefit with sensitivity identification. As shown in Section 5, in every models, zero-shot dropping can obtain at least 44% of blocks as SPD with little sacrifice of accuracy.

### 4.2.3 SENSITIVE BLOCKS: SPD AWARE BLOCK-TO-BLOCK DISTILLATION

Sensitive blocks exhibit larger effect on quality degradation compared to in-sensitive blocks. To further increase the optimization objectives and recover the associated performance degradation by dropping sync-point in sensitive block, we obtain SPD specific parameter by adopting the concept of block-to-block distillation. Block-to-block distillation is a low-cost training method that involves training only the specific sensitive block with SPD. The training objective of distillation is shown in Equation 1. We set teacher block as TP block and student block as SPD block. For the data used in tuning, we utilize same calibration data used in sensitivity identification step in Section 4.2.1. This calibration data passes through consecutive TP blocks of the model by the block which distillation will be conducted. To conduct distillation, we forward this hidden representation ($x$) to each teacher and student block and apply outputs to mean squared error (MSE) loss. Note that the parameter of SPD block ($\theta_{spd}$) is initialized from TP block ($\theta$). Since SPD and TP are execution methods within the system, they originally use the same model parameters. However, to make the separate weights aware of elminated communcation, parameters for SPD are newly initialized and used separately.

$$\underset{\theta_{spd}}{\arg\min} \ \mathrm{MSE}(\mathrm{SPD}(\theta_{spd}, x), \mathrm{TP}(\theta, x)) \tag{1}$$

### 4.2.4 EXTREMELY SENSITIVE BLOCKS: ATTENTION HEAD GROUPING ENHANCEMENT FOR SPD AWARE BLOCK-TO-BLOCK DISTILLATION

Beyond the recovery of block-to-block distillation on sensitive blocks, a few number of blocks show sharp quality degradation. We define these blocks as extremely sensitive blocks and introduce a novel SPD aware initialization for block-to-block distillation. As the sync-points are removed, the model partitions are isolated from each other, preventing mutual access. This makes a decoder block as if it is a combination of parallel and independent mini decoder blocks. In this circumstance, a self attention fragment cannot access any MLPs in other parallel devices and also MLPs are unable to access self attention output in other parallel devices, resulting in inevitable information loss. To ensure that these parallel architectures operate as close as the original structure, it is important to make attention heads evenly distributed based on functionality following the sparse nature of head activated differently (Liu et al., 2023) and redundancy of head showing similar behaviors (Agarwal et al., 2024) on in-context. To reflect these in-context properties to out-context as much as possible, we utilize calibration data and obtain attention score ($\sigma$) as a metric of the head functionality.

**Head scattering** In the self attention, the set of query (Q), key (K) and value (V) associated with each head can be defined as $A = \{< Q_1, K_1, V_1 >, < Q_2, K_2, V_2 >, \cdots, < Q_N, K_N, V_N >\}$ where $N$ is number of heads. The goal of head scattering is finding the set of heads showing the even distribution of attention score ($\sigma(Q_i, K_i)$) on each device. By defining $A_i \subset A$ where $n(A_i) = N/number\_of\_devices$, the objective of head scattering is defined in Equation 2. We achieve the objective of finding even distribution based on head functionality by maximizing sum of distance on clustering algorithm which originally utilize opposite metric. Here, for the distance, attention scores of each sequences as a high dimension vector are utilized with euclidean distance ($d$).

$$\underset{A_i}{\arg\max} \sum_{j=1}^{n(A_i)} \sum_{k=j+1}^{n(A_i)} d(\sigma(Q_{A_{i,j}}, K_{A_{i,j}}), \sigma(Q_{A_{i,k}}, K_{A_{i,k}})), \quad \text{where} \quad A_i \subset A \tag{2}$$

**MLP matching** After getting the scattered clusters of attention heads, matching $A_i$ with proper MLP partition should be conducted to search complete parallel independent architecture which operate close to the original structure. We found that the norm of MLP output before adding residual connection is well fit indicator to maximize the impact of scattered head subset. Therefore, we compare the norm of all the matching combinations and pick the best maximum case as matching result. By defining MLP partition of a device as $MLP_m$ and a matching combinations as $MC$ and its universal set as $MC_{all}$, the objective of MLP matching is defined as Equation 3.

$$\arg\max_{MC} \sum_{<A_i, MLP_m>}^{MC} Norm(MLP_m(A_i)), \quad \text{where} \quad MC \in MC_{all} \quad (3)$$

After determining the optimal $A_i$ and $MC$, the hidden representation of each head should be physically located on the device designated by $MC$. Figure 4 illustrates the example of SPD with best $A_i$ and $MC$. To align the assignment with the static behavior of the system in SPD, we reorder the columns of the query, key, and value linear layer weights ($W_Q$, $W_K$, $W_V$) based on their head-specific partitions ($W_{Qh}$, $W_{Kh}$, $W_{Vh}$, where $h$ denotes the head index). Similarly, the row order of output linear layer weight ($W_O$) based on head partitions ($W_{Oh}$) are reordered. This reordering ensures that the hidden representations are distributed in the order of $MC$, allowing the heads in $A_i$ to reside on the same parallel device. As a result, a group of scattered head subset and MLP partition is assigned to a single device, referred to as head grouping. Applying block-to-block distillation after head grouping further enhances accuracy recovery in the extremely sensitive blocks.

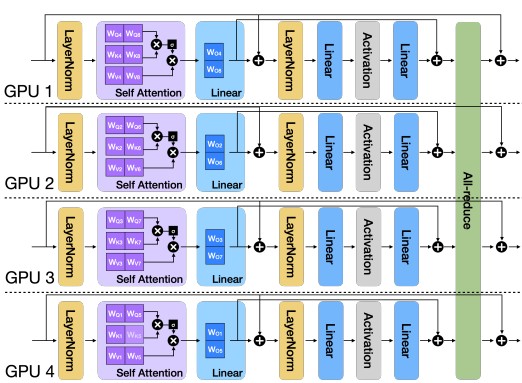

$\{A_1, A_2, A_3, A_4\} =$
$\{\{<Q_2, K_2, V_2>, <Q_6, K_6, V_6>\}, \{<Q_1, K_1, V_1>, <Q_5, K_5, V_5>\},$
$\{<Q_3, K_3, V_3>, <Q_7, K_7, V_7>\}, \{<Q_4, K_4, V_4>, <Q_8, K_8, V_8>\}\}$
$MC = \{<A_4, MLP_1>, <A_1, MLP_2>, <A_3, MLP_3>, <A_2, MLP_4>\}$

Figure 4: SPD block structure without bias having 8-heads on 4-GPUs case with given $A_i$ and $MC$.

## 5 EXPERIMENTS

### 5.1 SETUP

**Models** We conduct experiments on LLaMA2 (7B, 13B and 70B) (Touvron et al., 2023b) and OPT (6.7B, 13B, 30B and 66B) (Zhang et al., 2022). We apply 8-GPUs case and 4-GPUs case settings for all the models except LLaMA2-70B, OPT-30B and 66B which apply 8-GPUs case setting only.

**Calibration data** From WikiText2 (Merity et al., 2016) training dataset, randomly selected 128-samples consist of tokens with sequence length of 2048 are used by following existing work (Shao et al., 2024). Each one sample of calibration data is utilized as mini batch for distillation.

**Settings** For all models except larger models (LLaMA2-70B, OPT-30B and OPT-66B), we use $\tau_1$ as 0.05 and $\tau_2$ as 10. For larger models, we use $\tau_1$ as 0.02 and $\tau_2$ as 10. In block-to-block distillation on sensitive blocks, learning rate is used as $5 \times 10^{-5}$ for LLaMA2 and $1 \times 10^{-6}$ for OPT. 10-epochs of distillation is conducted with 1-epoch as utilizing whole 128-samples of calibration data.

**Evaluation data** We evaluate accuracy of our optimization method to zero-shot tasks (ARC (Clark et al., 2018), HellaSwag (Zellers et al., 2019), LAMBADA (Paperno et al., 2016), PIQA (Bisk et al., 2020), SciQ (Welbl et al., 2017), and WinoGrande (Sakaguchi et al., 2020)) by averaging all the results and MMLU tasks (Hendrycks et al., 2021).

### 5.2 SENSITIVITY IDENTIFICATION

Figure 5 shows the block-wise sync sensitivity identification result of the blocks in LLaMA2 and OPT models. For all models, the percentage of in-sensitive blocks (yellow bar) indicate that the

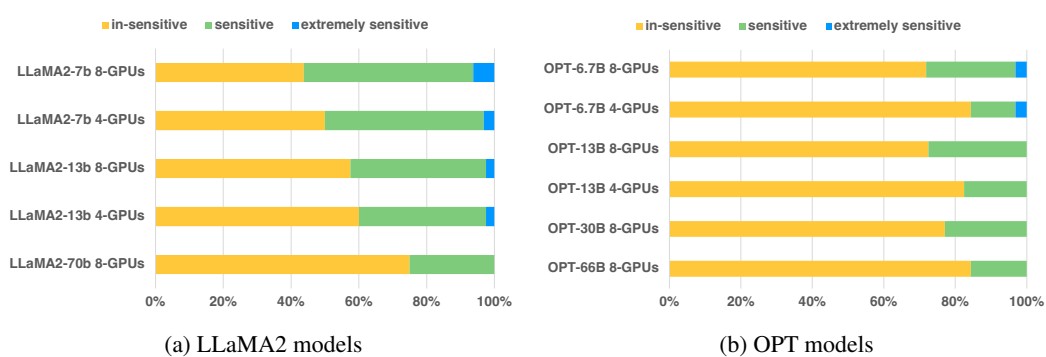

Figure 5: Block-wise sync sensitivity identification result for LLaMA2 and OPT models.

same amount of blocks can be used as SPD with ignorable accuracy drop (less than 1% on zero-shot tasks). This can be achieved in zero-shot manner (detailed results are described in Section 5.3). The percentage of in-sensitive blocks increases when the model size gets larger (75% in LLaMA2-70B 8-GPUs and 84% in OPT-66B 8-GPUs). Overall, LLaMA2 models show higher sensitivity compared to OPT models. LLaMA2-7B 8-GPUs model is available with zero-shot drop of 44% while entire OPT models are available with dropping 70% of blocks. Extremely sensitive blocks are shown only in smaller models (LLaMA2-7B, 13B and OPT-6.7B) with the amount of one or two blocks.

## 5.3 SENSITIVITY BASED SYNC-POINT DROP

Figure 6 shows the SPD results of LLaMA2 models on zero-shot tasks. After the amount of target SPD blocks exceeds in-sensitive boundary, zero-shot dropping (ZS) shows large accuracy drop (over 1%) in all models and cases. Block-to-block distillation with ZS (ZS+B2B) successfully recovers large amount of accuracy in sensitive block region, especially giving larger amount on smaller models (+28% on 13B 8-GPUs and +20% on 7B 4-GPUs on 100% SPD). Furthermore, smaller models having extremely sensitive blocks show further accuracy recovery from B2B (+3% on 7B 8-GPUs and +2% on 13B 4-GPUs on 100% SPD) with adding head grouping enhancement (ZS+B2B+HG). Similar tendencies are also appeared on MMLU results as in Figure 7.

Figure 8 shows the SPD results of OPT models on zero-shot tasks. OPT models show less drop compared to LLaMA2 models possibly due to high redundancy (Liu et al., 2023; Agarwal et al., 2024). Models except 1.3B show maximum 1.3% degradation regardless of sensitivity of block. Therefore, results in OPT with ZS+B2B and ZS+B2B+HG shows no large improvements since it already have less drop only with ZS. However, in OPT-6.7B, when the drop occurs in ZS, ZS+B2b and ZS+B2B+HG gives recovered accuracy (+2.8% in 8-GPUs and +2% in 4-GPUs on 100% SPD).

Overall the proposed SPD effectively alleviates sync-point bottleneck while minimizing accuracy degradation. This shows that SPD gives both moderate optimization with no performance degradation and better trade-off between larger optimization and performance leading to scalable solution.

## 5.4 EFFECTS OF DESIGN CHOICE IN BLOCK DESIGN

Table 2: SPD MLP output design choice Wiki-Text2 perplexity on block without bias in linear layer (SPD is only on 1st block of the model).

Table 3: SPD MLP output design choice Wiki-Text2 perplexity on block with bias in linear layer (SPD is only on 1st block of the model).

| Attention output residual add ($Y_i$) | PPL ($\downarrow$) |
|---|---|
| LLaMA2-7B no SPD | 5.47 |
| Before MLP all-reduce | 10.65 |
| After MLP all-reduce | 177.69 |

| Bias residual add ($b$) | PPL ($\downarrow$) |
|---|---|
| OPT-6.7B no SPD | 10.86 |
| Before MLP all-reduce | 332.60 |
| After MLP all-reduce | 13.07 |

Section 4.1 shows that the tensor parallelism block system is not compatible with lack of communication and this makes several design choices on block structure. Table 2 and 3 show quality

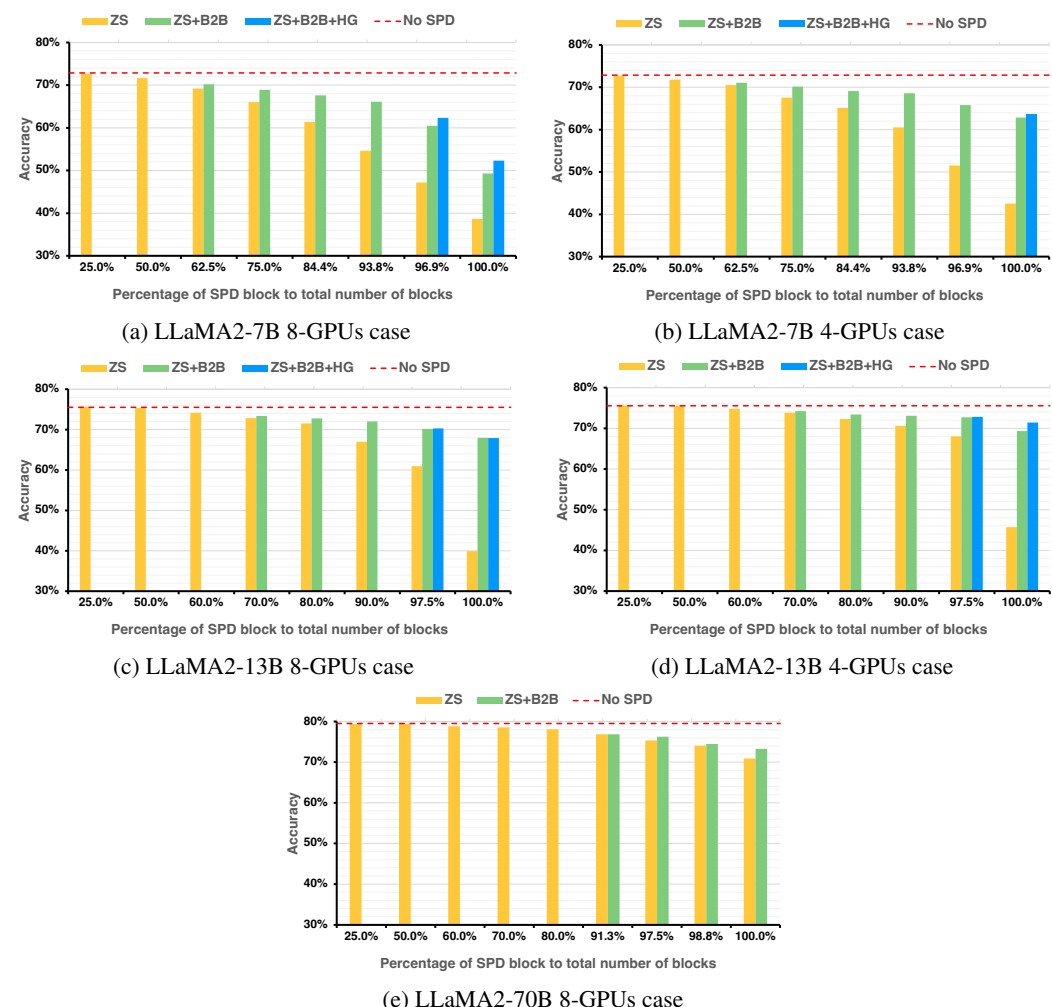

Figure 6: LLaMA2 average accuracy results on zero-shot tasks ('ZS' represents applying zero-shot dropping to all blocks. 'ZS+B2B' represents applying zero-shot dropping on in-sensitive blocks and block-to-block distillation to the other blocks. 'ZS+B2B+HG' is applying zero-shot dropping on in-sensitive blocks and block-to-block distillation to sensitive blocks and block-to-block distillation with head grouping enhancement to the other blocks which is extremely sensitive blocks).

degradation per design choice on MLP output. Whether the targeted residual connections on each table use collective communication or not will be determined by the addition point (before and after MLP all-reduce). The results show that using collective communication on attention output residual (Table 2) and not using it on bias (Table 3) are the proper choice of residual addition point design selections as in Figure 2 which minimizes negative effect from SPD.

## 6  CONCLUSION

In this paper, we present Sync-Point Drop (SPD), a novel optimization technique improving the latency of LLMs on distributed inference environment by reducing the communication overhead in tensor parallelism with model-side solutions. By identifying regions of communication redundancy and selectively omitting synchronization on attention outputs, SPD enables efficient deployment across multiple computing units with little compromising model performance. To extend the range of SPD with little degradation, our block-wise sync sensitivity analysis allows us to target only blocks that experience accuracy degradation with a low-cost block-to-block distillation process, ensuring minimal quality drop. For blocks largely impacted by syncing, we introduce block-to-block

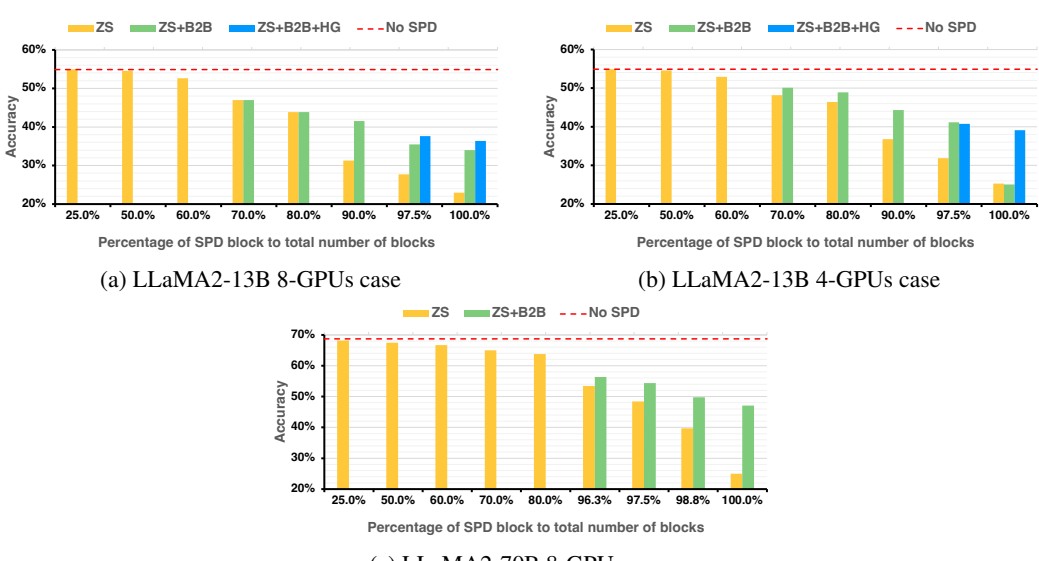

Figure 7: LLaMA2 accuracy results on MMLU tasks (Notations are same as in Figure 6).

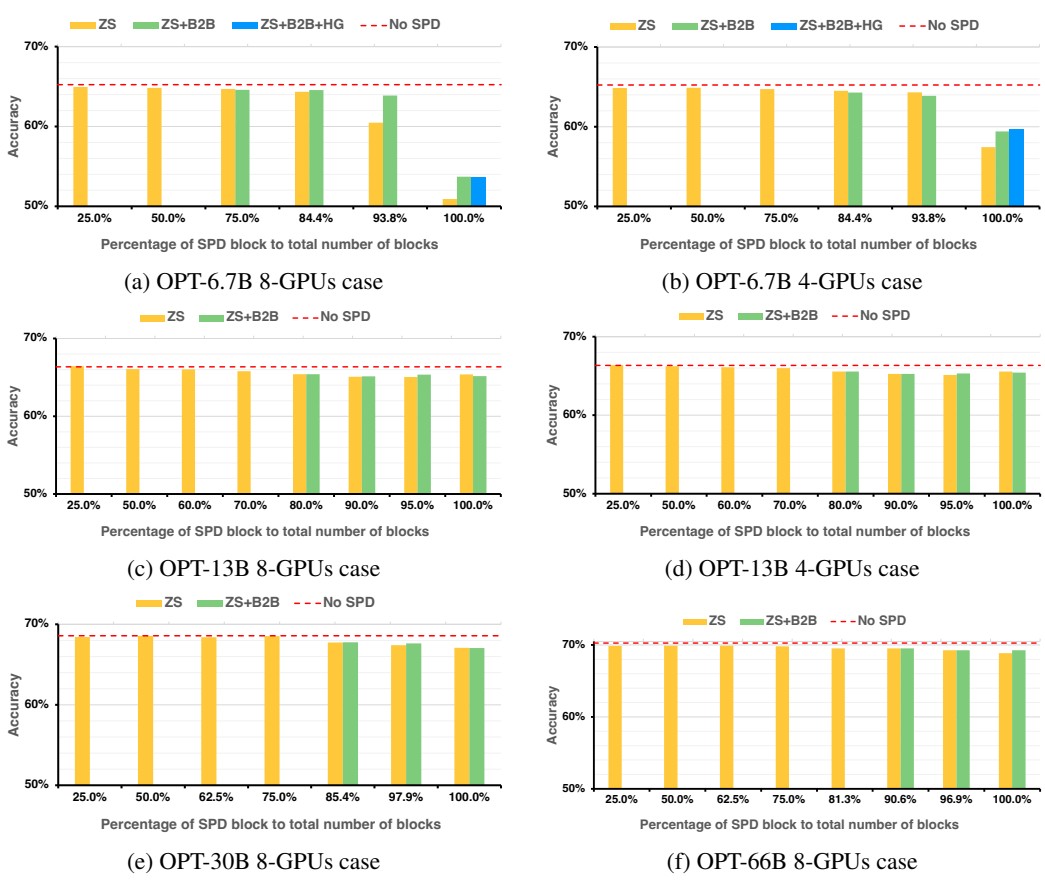

Figure 8: OPT average accuracy results on zero-shot tasks (Notations are same as in Figure 6).

distillation and attention head grouping enhancements to maximize the recovery from dropped quality. Our experiments show that SPD offers improved latency with minimum quality loss in all budgets which enable scalable solution for distributed environments.

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
