# OpenReview forum: "SPD: Sync-Point Drop for efficient tensor parallelism of Large Language Models"
_ICLR.cc/2025/Conference — Submitted to ICLR 2025_

### Official Review · Reviewer_gSyN · 2024-10-29

**Soundness:** 2
**Presentation:** 3
**Contribution:** 4
**Rating:** 5
**Confidence:** 5

**Summary:**

This paper introduced SPD, a special TP inference block aiming to reduce communication overheads. The authors proposed a block design that allows execution to proceed without communication redundancy. Also, the authors employed a low-cost distillation targeting blocks giving quality degradation. The authors claimed to alleviate communication bottlenecks while minimizing accuracy degradation during LLM inference and offered a scalable solution for distributed environments.

**Strengths:**

1. SPD removed all-reduce communication after self-attention which could help reduce TP's communication issue. It is known that TP is communication heavy parallelism.
2. Although SPD will degradate model performance, the authors showcases that their ZS+B2B and ZS+B2B+HG could help to alleviate this degradation even with a high percentage of SPD changed blocks. It is performing well even with 91.3% of SPD block in the LLaMA2-70B 8-GPUs case.
3. Based on existing experiment results, larger models with less GPU number are less affected by SPD changes. This technique could be helpful for both LLM academia and the LLM industry in improving inference efficiency.

**Weaknesses:**

1. There is not a table showing the latency differences on different models, which would be great also critical to showcase the efficiency improvement.
2. Latency differences in Table 1 only show latency with Llama2-7B, it would be better and necessary to show latency differences of other model sizes as well.
3. Experiments are not well designed. There is no analysis of latency and accuracy comparison while controlling the model sizes and GPU numbers. Showing latency improvement and accuracy recovery chart in a separate manner does not help to prove this paper's claim.

**Questions:**

1. It is common to keep all TP communications inside a single node as much as possible, but Table 1 compared a intra-node situation. Could you also provide how are these nodes connected?
2. Could you also provide a comparison of latency difference on different models? Accuracies are compared in Figure 6 but it is not so straightforward on the latency side.
3. Is it possible to test on larger models like the 405B llama model? Seems like larger models are less sensitive to the SPD changes.
4. Are there any experiments on removing the other all-reduce op after MLP? Would it also be helpful?

---

### Official Review · Reviewer_EqMR · 2024-10-30

**Soundness:** 3
**Presentation:** 3
**Contribution:** 3
**Rating:** 5
**Confidence:** 4

**Summary:**

The authors propose SPD, an communication optimization method that reduces communication overhead by dropping synchronization points within transformer layers. To maintain performance, SPD employs block-to-block distillation and attention head-grouping on sensitive blocks. Experimental results indicate minimal performance losses across various models and datasets, demonstrating the method's effectiveness.

**Strengths:**

- The work identifies that the majority of transformer layers are insensitive to synchronnization dropping for Tensor Parallelism (TP).
- Dropping synchronization points in TP effectively alleviates the communication overhead, thereby accelerating the inference of transformers.
- Experimental results show that SPD incurs minimal performance degradation when synchronization points are dropped for more than half  of layers.

**Weaknesses:**

- The analysis of communication overheads of TP across different settings is lacking, which could clarify the motivation for the proposed method.
- There is no comparison of communication costs before and after applying SPD. Additionally, the experimental section lacks baseline comparisons. For example, communication in TP could be overlapped with computations.
- Latencies after applying SPD on LLaMA2-13B and LLaMa2-70B are not provided. Only per-block latencies for SPD on LLaMA2-7B are presented in Table 1.

**Questions:**

- Could you provide overall latencies across different models for SPD and TP for better comparison?
- The identificantion of sensitivity for blocks relies on a calibration dataset. Does the sensitivity of a block vary with different types of data?

---

### Official Review · Reviewer_NF7u · 2024-11-03

**Soundness:** 2
**Presentation:** 1
**Contribution:** 2
**Rating:** 3
**Confidence:** 4

**Summary:**

The authors propose SPD a mechanism in which they can remove certain synchronization in tensor parallel.

To reduce loss in accuracy the authors come up with certain additional ideas which require some fine-tuning.

**Strengths:**

- The paper tackles a highly relevant problem of minimizing latency

- The authors have spent quite an amount of effort trying to minimize the accuracy loss

**Weaknesses:**

- The paper fails to do end to end comparison on run time.

- I think the paper at several places is hard to follow because of writing.

**Questions:**

“The inefficiency of large language models, which emerged with significant impact” ([“SPD”, p. 2]-> Grammar

“To make the lowest bandwidth requirement between devices as possible” ([“SPD”, p. 3]-> Grammar

“Block-to-block distillation is a low-cost training method that involves training only the specific sensitive block with SPD” ([“SPD”, p. 6]-> At this point what is SPD, I think the terminology is being used pretty excessively. Is SPD a block, is SPD a method is SPD the insight, there has to be some sort of clean demarcation.

“We found that the norm of MLP output before adding residual connection is well fit indicator to maximize the impact of scattered head subset” ([“SPD”, p. 7]-> Eval on this and why is this a reasonable metric.

“Figure 4: SPD block structure without bias having 8-heads on 4-GPUs case with given Ai and M C.” ([“SPD”, p. 7](zotero://select/library/items/W6D6RQAL)) ([pdf](zotero://open-pdf/library/items/FVKEHWVW?page=7)) \-> Why do you still need All-reduce after partitioning, if the very sensitive blocks need synchronization then why do the partitioning, rather you can an gather scatter once.

“Figure 5: Block-wise sync sensitivity identification result for LLaMA2 and OPT models.” ([“SPD”, p. 8](zotero://select/library/items/W6D6RQAL)) ([pdf](zotero://open-pdf/library/items/FVKEHWVW?page=8)) \-> Can you provide details on which blocks were determined overly sensitive

“8: τ1, τ2 ← sensitivity thresholds” ([“SPD”, p. 5]-> How do you come with these sensitivity thresholds.

“To reflect these in-context properties to out-context as much as possible, we utilize calibration data and obtain attention score (σ) as a metric of the head functionality.” ([“SPD”, p. 6]

“After the amount of target SPD blocks exceeds in-sensitive boundary,” ([“SPD”, p. 8]> What does this mean ?

What about end to end time improvement numbers ?

In most cases you can use SPD to drop around 60% layers without any accuracy loss ? Do you think there is significant need for the rest of the techniques proposed.

---

### Official Review · Reviewer_2KQ3 · 2024-11-03

**Soundness:** 2
**Presentation:** 3
**Contribution:** 2
**Rating:** 3
**Confidence:** 3

**Summary:**

The paper introduces Sync-Point Drop (SPD), an optimization technique aimed at reducing communication overheads in distributed inference for large language models (LLMs) by minimizing synchronization on attention outputs within tensor parallelism. It proposes a block design that minimizes communication while preserving model accuracy, differentiating block sensitivity to communication (in-sensitive, sensitive, extremely sensitive) for optimal SPD application.

**Strengths:**

1. It proposes a block design that minimizes communication while preserving model accuracy, differentiating block sensitivity to communication (in-sensitive, sensitive, extremely sensitive) for optimal SPD application.
2. The method identifies sensitivity across model blocks, applying SPD only where communication reduction does not degrade performance.
3. The paper employs zero-shot dropping for in-sensitive blocks, block-to-block distillation for sensitive blocks, and head grouping enhancements for extremely sensitive blocks, to balance latency reduction with minimal accuracy loss.

**Weaknesses:**

1. While SPD is validated across several models, its effectiveness may vary with different architectures or tasks outside those tested, potentially limiting its broader applicability.
2. The paper’s approach requires fine-tuned block sensitivity analysis and specific design adjustments for each block type, which may complicate implementation in practical, large-scale deployments.
3. Although SPD aims to minimize accuracy loss, the approach relies on careful tuning of sensitivity thresholds, which, if not optimized correctly, could still lead to notable performance degradation.

**Questions:**

1. The main purpose of this paper is to reduce communication overheads in tensor parallelism. Why not provide the final speedup after removing one all reduce? Or at least provide the theoretical speedup？
2. For modern GPUs, like DGX H100, the communication is already very fast, the speedup of removing one allreduce may be very limited after doing such a complicated process.
3. Do we have a method to determine Percentage of SPD block to total number of blocks? Or users have to try different values?

---

### Comment · Area_Chair_a3ba · 2024-11-21
**No author response yet**

Dear Submission12793 Authors,

ICLR encourages authors and reviewers to engage in asynchronous discussion up to the 26th Nov deadline. It would be good if you can post your responses to the reviews soon.

---

### Meta-Review · Area_Chair_a3ba · 2024-12-09

**Metareview:**

The paper proposes a system to reduce communication in tensor parallel model training, by opportunistically dropping synchronization on certain attention outputs, thus reducing communication latency.

Reviewers agreed that the communication latency problem is well-motivated. However, the initial version of the paper lacked latency experiments, and multiple reviewers pointed out other issues with the experimental design. The author rebuttal provided some new experiment results, but revealed that the performance gains of the method were not convincing: for example, a 6% speedup with 4% accuracy drop.

**Additional Comments On Reviewer Discussion:**

Reviewers requested new experiments on larger models such as LLaMa2-7b. While some results were provided by the author rebuttal, the speedups reported were not convincing: for example, a 6% speedup with a 4% accuracy drop. The system needs further optimization to show a convincing speedup that clearly outweighs the loss in accuracy.

---

### Decision · Program_Chairs · 2025-01-22

Reject